# Cytokines interferon−γ− inducible protein 10 and granulocyte−macrophage colony−stimulating factor are associated with psychiatric symptoms in opioid−dependent patients: A cross− sectional study

Kristin Nygård−Odeh[1,2]* , Hedda Soløy−Nilsen[1,2] , Magnhild Gangsøy−Kristiansen[2], Ole Lars Brekke[1,2], Tom Eirik Mollnes[3,4], Michael Berk[5,6]

1 Nordland Hospital Trust, Bodø, Norway, 2 Institute of Clinical Medicine, UIT − The Arctic University of Norway, Tromsø, Norway, 3 Research Laboratory, Nordland Hospital Trust, Bodø, Norway, 4 Department of Immunology, Oslo University Hospital and University of Oslo, Oslo, Norway, 5 Deakin University, IMPACT – the Institute for Mental and Physical Health and Clinical Translation, School of Medicine, Barwon Health, Geelong, Australia, 6 Orygen, The National Centre of Excellence in Youth Mental Health, Centre for Youth Mental Health, Florey Institute for Neuroscience and Mental Health and the Department of Psychiatry, The University of Melbourne, Melbourne, Australia

☯ These authors contributed equally to this work.
* Kristin.Nygard-Odeh@nordlandssykehuset.no

## Abstract

### Background

Psychiatric disorders and chronic hepatitis virus C infection are known to alter blood cytokines levels. However, little is known about the association between cytokines and psychiatric symptoms in patients with chronic hepatitis C virus infection. This study aimed at exploring this association. Moreover, since nearly half of the patients receive opioid maintenance treatment, we also investigated if long−term opioid treatment had any impact on these associations.

### Methods

We conducted a cross−sectional study on 120 outpatients referred for antiviral hepatitis C treatment. Serum level of 27 cytokines was measured using multiplex technology, and psychiatric symptom clusters were assessed using the Symptoms Check−List−90−R. Data on confounding factors including age, gender, weight, height, current medication and smoking habits were collected. Multiple linear regression analysis was performed to examine associations, adjusting for confounding factors.

### Results

After adjusting for the most commonly known confounding factors, IP−10 and GM−CSF were negatively associated with depression, and GM−CSF was negatively

**Data availability statement:** Data will be made available upon direct request to the following institution: Nordland hospital trust Department of research P.o. box 1480 N-8092 Bodø Norway forskning@nordlandssykehuset.no.

**Funding:** The study was in part funded by research grant from Northern Norway regional health authority (reference: RUS1303-16). The funders had no role in the study design, data collection and analysis, decision to publish, or preparation of the manuscript.

associated with phobic anxiety. Subgroup analyses revealed that these associations were present only in patients receiving opioid maintenance treatment, as demonstrated by repeated regression analysis.

## Conclusions

In patients with chronic hepatitis C viral infection, only IP-10 and GM-CSF were negatively associated with self-reported psychiatric symptom clusters. These associations were observed exclusively in patients receiving opioid maintenance treatment. Our study contributes to others investigations pointing to a possible immune dampening caused by long-term opioid treatment.

## Introduction

Emerging evidence of immunological activation in psychiatric disorders has spurred research into the association between cytokines and various psychiatric disorders [1,2]. For instance, tumour necrosis factor-alpha (TNF) and interferon-gamma (IFN-γ) have been linked to generalized anxiety [2], while interleukin-6 (IL-6) are elevated in patients with major depressive disorder, suggesting that cytokines could serve as potential diagnostic biomarkers [1]. Similar to their use as diagnostic markers and therapeutic targets in inflammatory disorders [3], identifying associations between cytokines and psychiatric disorders may have important clinical implications. However, relying solely on diagnosis-based associations poses limitations, as psychiatric diagnoses are often co-occurring [4]. This leads to a potential overemphasis on one diagnosis while neglecting others. Moreover, inter-rater diagnostic variability challenges diagnostic accuracy [5]. Since symptoms frequently overlap and co-occur across many psychiatric diagnoses [6], investigating the relationship between cytokines and specific symptoms may offer a more accurate and clinically useful approach. Previous studies have explored associations between cytokines and symptoms such as anhedonia [7], fatigue [8], psychotic symptoms [9], agitation [10] and hostility [6].

Opioid maintenance treatment (OMT) is a well-established intervention for opioid dependence, involving the use of long-acting opioids. Notably, 43 percent of these patients have chronic hepatitis C viral infection (HCV) [11]. Both long-acting opioids [12–14] and chronic HCV [15,16] impact the immune system and alter cytokine levels. For example, 12 weeks of methadone treatment has been shown to reduce the levels of cytokines transforming growth factor-beta (TGF -β) and brain-derived neurotrophic factor (BDNF) [12], while prolonged methadone use (median 23.6 months) is associated with higher levels of interleukin-1 beta (IL-1β), IL-6 and interleukin-8 (IL-8) compared to healthy controls [13]. Similarly, chronic HCV infection leads to elevated levels of cytokines, including interleukin-1 alpha (IL-1α), IL-1β, IL-6 and IL-8 [17]. As chronic HCV progresses to liver cirrhosis, levels of IL-1α, interleukin-2 receptor (IL-2R), interleukin-12 (IL-12), interleukin-18 (IL-18) and C-X-C motif chemokine ligand 9 (CXCL9) increase, suggesting an association with disease

progression [16]. Fifty percent of both HCV [18]− and OMT−patients [19] have psychiatric symptoms. This comorbidity is associated with poorer adherence to treatment in both groups [20,21]. Investigating the relationship between cytokines and psychiatric symptoms in this context may help identify diagnostic markers and pave the way for individualized, symptom−oriented treatment [22], ultimately improving treatment outcomes. Therefore, the aim of this study was to investigate the relationship between peripherally circulating cytokines and psychiatric symptoms in patients with chronic HCV infection. Additionally, we wanted to study the impact of opioid maintenance treatment on these associations by comparing non−OMT with OMT patients.

## Materials and methods

### Design, recruitment and participants

This study used data from a previously published cross−sectional study [23]. Patients referred for antiviral HCV treatment were recruited from the Department of infectious diseases at the Nordland Hospital Trust, Bodø, Norway in the period of April 23rd 2013−December 12th 2019. On the day of their appointment, a study nurse informed the patients about the study and invited them to participate. Interested individuals were provided with detailed verbal and written information by the main investigator (first author), and were asked to provide written consent [23]. Exclusion criteria included patients who did not understanding the Norwegian language, had obvious cognitive deficits, negative HCV−RNA, failure to obtain written consent and failure to provide blood samples. Of the 135 screened patients, 15 were excluded due to negative HCV−PCR (n = 9), failure to provide written consent (n = 4) and failure to provide blood−samples (n = 2). In the final sample of 120 patients, 53 were on OMT, and 67 patients were not [23]. The study was approved by the Regional ethics committee (notification 2015/1808/REK Nord and 2011/2024/REK Nord), and adhered to the principles of the Declaration of Helsinki.

### Data collection

Upon recruitment, participants completed the self−administered form, the Symptoms Check−List−90−Revised (SCL−90−R). The SCL−90−R is a validated questionnaire investigating 90 items screening for psychological distress during the last week, each of which rated on a five−point scale of distress [24]. The 90 items are grouped into 9 subscales/clusters (somatization, obsessive−compulsive behaviour, interpersonal sensitivity, depression, anxiety, hostility, phobic anxiety, paranoid ideation, psychoticism) that have good reliability and replicability [25]. The scale ranges from "not at all" (0) to "extremely" (4). Mean scores across the clusters were calculated. Patients were interviewed by the main investigator using a modified adaptation of the multidimensional European Addiction Severity Index (Europ−ASI) questionnaire [26]. The version included sections general information, physical health, economy, education and employment−status and alcohol and other drugs−use. Interviewer's severity assessment and reliance assessment were omitted. Data were collected on OMT drug type and dosage, gender, age, weight, height, body mass index (BMI = kg/m$^2$), smoking habits, current medication, allergies, medical history and ongoing withdrawal symptoms [23].

### Blood sampling and analyses

Blood samples were drawn by the study nurse on the day of inclusion between 11:00 a.m. and 2:50 p.m. [23]. Biochemical measures were performed at the Central laboratory, Department of Laboratory Medicine, Nordland Hospital Trust. For measurement of cytokines, blood was withdrawn in Vacuette serum tubes, left for 30 minutes before centrifugation 10 minutes at 2300 x g. Serum (2 x 1 mL) was stored in Matrix tubes on ice up to 2 hours before freezing at −80º C [23].

The cytokine analysis was performed by multiplex technology with a predefined kit Bio−Plex Human Cytokine 27−Plex Panel (Bio−Rad Laboratories Inc., Hercules, CA) according to the instructions of the manufacturer [23] The assay detected the following interleukins, chemokines, and growth factors: tumour necrosis factor (TNF), interferon

(IFN)−gamma, IL-1β, IL-1 receptor antagonist (IL-1ra), IL-2, IL-4, IL-5, IL-6, IL-7, IL-8 (C−X−C motif chemokine ligand 8; CXCL8), IL-9, IL−10, IL-12, IL-13, IL-15, IL−17, monocyte chemotactic protein (MCP−1/CCL2), interferon−γ−inducible protein 10 (IP−10/C−X−CL chemokine 10; CXCL10), eotaxin−1 (C−C motif chemokine ligand 11; CCL11), macrophage inflammatory protein−1α (MIP−1α/ CCL3), macrophage inflammatory protein−1−β (MIP−1β/ CCL4), regulated upon acti-vation T cell expressed and secreted (RANTES), granulocyte macrophage colony stimulating factor (GM−CSF), vascular endothelial growth factor (VEGF), basic fibroblast growth factor (bFGF), granulocyte−colony stimulating factor (G−CSF) and platelet derived growth factor−BB (PDGF−BB) [23].

## Statistical analyses

Tests for cytokine level−normality was performed using Q−Q plots and found to be normal with $lg_{10}$ transformation. Extreme outliers defined as lying outside the third quartile + 3 x interquartile range were identified by boxplots. Their impact was tested using independent samples t−test before and after their removal. No impact was found. Cytokine values below lower limit of detection (LLOD) were assigned a value randomly drawn by Excel between LLOD and zero [23]. The cytokines with more than 50% randomly drawn numbers were excluded from further analyses, excluding β−FGF, IL−7, IL−10, and G−CSF. Cytokines in which more than 50% had values below lower standard value were also excluded from further analysis, eliminating IL−5, IL−10 and IL−12. Multiple linear regression analysis was performed with cytokines as dependent variables and symptom clusters and confounding factors (age, gender, BMI, smoking) as independent vari-ables. Hierarchical linear regression analysis was used in the further testing of IP−10 and GM− CSF. Testing with variance inflation factor below 10, showed no problems with collinearity [23]. Comparisons of the variables between the groups were performed using Mann−Whitney U test for continuous vs. binary variables and non−normal distributed variables, and chi−square tests for categorical variables. P−values <0.05 were deemed statistically significant [23]. Statistical analyses were performed using IBM SPSS Statistics viewer version 29.0.1.0.

## Inclusivity in global research

Additional information regarding the ethical, cultural and scientific considerations specific to inclusivity in global research is included in the Supporting Information (SX Checklist).

## Results

In the 120 included patients, the median age was 45 years, 39 were women and 108 were Norwegian nationals. Fifty−three were on OMT (Table 1).

Data on smoking were missing for 30 patients, while data on ongoing psychotropic drug administration were missing for 58 patients. Cytokines values presented in Table 2 are the non−$lg_{10}$ transformed values.

From a previous study, cytokine levels in a group of healthy controls were used to validate our levels and can be found in the Supporting information (S1 Table) [27]. Most cytokine levels were low both in in the participants and in the healthy controls.

The multiple linear regression analysis with cytokines as dependent variable showed significant negative associa-tions in two symptom clusters after adjusting for age, gender, BMI and smoking (Table 3): IP−10 (β = −0.23, p = 0.04) and GM−CSF (β = −0.26, p = 0.04) were negatively associated with depression, whereas GM−CSF (β = −0.26, p = 0.03) was negatively associated with phobic anxiety. Neither of these two cytokines had any association with HCV−infection status (IP−10: HCV−RNA: β = −0.02, p = 0.88. Ratio ASAT/ALAT: β = 0.17, p = 0.14. GM−CSF: HCV−RNA: β = −0.01, p = 0.91. Ratio ASAT/ALAT: β = 0.05, p = 0.64).

To investigate if opioids had any impact on these associations, participants were stratified into non−OMT and OMT groups. Sociodemographic characteristics differed minimally between the groups, with age being the only difference (Table 4). Median age in the non−OMT−group was 47 years and 42 years in the OMT group (p = 0.043).

**Table 1. Sociodemographic, clinical and psychometric characteristics of study participants (N = 120).**

| Variables | | |
|---|---|---|
| Sociodemographics | | |
| Female gender | N (%) | 39 (33) |
| Age (years) | Median (IQR) | 45 (35−54) |
| Norwegian nationality | N (%) | 108 (90) |
| Smoking | N (%) | 71 (59) |
| BMI | Median (IQR) | 25 (22−28) |
| Clinical characteristics | | |
| Virus load HCV RNA (IU/L *1000) | Median (IQR) | 891 (322.5−3266) |
| On OMT | N (%) | 53 (44) |
| On psychotropic drugs | N (%) | 34 (28) |
| Symptom cluster score in SCL−90−R | | |
| Depression | Median (IQR) | 1.23 (0.61−1.90) |
| Anxiety | Median (IQR) | 1.00 (0.40−1.70) |
| Somatization | Median (IQR) | 1.17 (0.70−1.90) |
| Psychoticism | Median (IQR) | 0.30 (0.10−0.65) |
| Phobic anxiety | Median (IQR) | 0.57 (0.14−1.43) |
| Paranoid ideation | Median (IQR) | 0.50 (0.83−1.17) |
| Hostility | Median (IQR) | 0.50 (0.50−1.33) |
| Interpersonal sensitivity | Median (IQR) | 0.67 (0.17−0.83) |
| Obsessive−compulsive | Median (IQR) | 1.25 (0.70−2.70) |

OMT, opioid maintenance treatment; BMI, body mass index; HCV, hepatitis C virus; SCL−90−R, Symptoms Check−List−90−Revised.

We ran a hierarchical regression analysis between the associated symptom cluster and cytokines that displayed significance in the linear regression analysis (Table 5). The significance was lost in the non−OMT group and remained significant throughout all the adjusting steps (age, gender, BMI and smoking) in the OMT−group with a negative association (depression/IP−10: β = −0.49, p = 0.001, depression/GM−CSF: β = −0.37, p = 0.02, phobic anxiety/GM−CSF: −0.32, p = 0.03).

## Discussion

In this study on patients with chronic HCV, we found that in the OMT patients IP−10 and GM−CSF were negatively associated with depression, and GM−CSF with phobic anxiety. No other association between nine symptom clusters and 27 cytokines were found in the entire study population.

IP−10, a member of the CXC chemokine family, mediates chemotaxis, apoptosis and angiostasis upon binding to the CXCR3 receptor [28]. It has been implicated in autoimmune diseases [29], infections [30] and tumours [31]. Studies on IP−10 and depressive symptoms show various and conflicting results. Elevated IP−10 levels have been observed in depressed adolescents compared to healthy controls [32], while no difference was reported in a geriatric population [33]. Consistent with our finding, IP−10 was lower in depressed patients with Takayasus arteritis compared to their non−depressed counterparts [34]. Studies investigating IP−10 as a potential prognostic therapeutic marker show mixed outcomes. While pharmaceutical antidepressant treatment did not have lasting impact on IP−10 levels [32,35], levels increased with symptom improvement following electroconvulsive treatment [36]. These results, coupled with our findings, suggest an inverse relationship between IP−10 levels and depressive symptoms, particularly in OMT patient.

GM−CSF drives the differentiation of progenitor cells into granulocytes and macrophages [37] and influences mature, differentiated cells [38] and non−hematopoietic cells [39]. It is implicated in inflammatory− and autoimmune disorders [40], and used in cancer immunotherapy [41]. Studies exploring the inflammatory hypothesis of depression, have shown higher

**Table 2. Median cytokines values of all study participants (N = 120).**

| Cytokine (pg/ml) | LLOD | N (%) <LLOD | Median (Q1−Q3) |
|---|---|---|---|
| TNF | 11 | 0 (0) | 50 (41−65) |
| IFN−γ | 0.36 | 12 (10) | 3.34 (1.40−8.33) |
| IL−1β | 0.04 | 0 (0) | 0.49 (0.32−1.00) |
| IL−1Ra | 18 | 0 (0) | 148 (121−218) |
| IL−2 | 0.84 | 0 (0) | 1.87 (1.17−2.66) |
| IL−4 | 0.36 | 2 (1.67) | 1.77 (1.07−2.60) |
| IL−6 | 0.40 | 0 (0) | 1.00 (1.00−1.00) |
| IL−8 | 1.00 | 0 (0) | 10 (5.00−12−00) |
| IL−9 | 1.48 | 0 (0) | 6.00 (5−9) |
| IL−13 | 0.24 | 0 (0) | 1.54 (0.97−2−21) |
| IL−15 | 6.32 | 12 (10) | 12 (4.41−28) |
| IL−17α | 1.24 | 0 (0) | 5.40 (3.70−7.18) |
| MCP−1 | 7.40 | 0 (0) | 35 (21−60) |
| IP−10 | 3.28 | 0 (0) | 235 (139−411) |
| Eotaxin | 0.40 | 0 (0) | 90 (63−127) |
| MIP−1α | 0.16 | 0 (0) | 1.56 (1.16−2.19) |
| MIP−1β | 1.48 | 0 (0) | 131 (114−141) |
| RANTES | 14 | 0 (0) | 25814 (13781−31949) |
| GM−CSF | 0.36 | 1 (0.83) | 0.84 (0.61−1.31) |
| VEGF | 12 | 12 (10) | 43 (36−54) |

LLOD, lower limit of detection; TNF, tumour necrosis factor; IFN−γ, interferon−gamma; IL-1β, interleukin−1beta; IL−1ra, interkeukin-1 receptor antago-nist; IL-2, interleukin−2; IL-4, interleukin−4; IL-5, interleukin−5; IL-6, interkeukin−6; IL-8, interleukin−8 (C−X−C motif chemokine ligand 8; CXCL8); IL-9, interleukin−9; IL-13, interleukin− 13; IL-15, interleukin−15; IL-17, monocyte chemotactic protein (MCP−1/CCL2); IP−10, interferon−γ−inducible protein 10 (C−X−CL chemokine 10; CXCL10); MIP−1α, macrophage inflammatory protein−1α (CCL3); MIP−1β, macrophage inflammatory protein−1−β (CCL4); RANTES, regulated upon activation T cell expressed and secreted; GM−CSF, granulocyte macrophage colony stimulating factor; VEGF, vascular endo-thelial growth factor.

baseline GM−CSF levels in depressed patients compared to healthy controls [42–44]. However, antidepressant treatment did not alter GM−CSF levels or have any association with clinical improvement [42–46]. The number of studies investigating the association between cytokines, depressive symptoms and OMT are scarce. One study measured levels of TNF, IL−1β and IL−2 [14]. As the patients transitioned from opioid addiction to treatment with either methadone or buprenorphine, levels of cytokines increased with subsequent lowering of depressive symptoms scores [14]. The negative association is consistent with our finding, albeit with different cytokines. These findings highlight the complexity of GM−CSF's role in depression and suggest that its relationship with symptoms may vary depending on the context, such as OMT.

While the number of studies linking GM−CSF to phobic anxiety are limited, associations with generalized anxiety disorders (GAD) have been documented, showing higher GM−CSF levels in GAD−patients compared to healthy controls [47]. This aligns with findings investigating GAD with other cytokines [48]. Our study is the first to suggest a potential relationship between GM−CSF and phobic anxiety in opioid dependent patients.

Both IP−10 and GM−CSF have pro−inflammatory properties [29,49], and the negative associations in a group of OMT patients with both infection and substance use as inflammatory drivers, were unexpected. This suggests that opioids may modulate these relationships in an immune−dampening way. The immunomodulatory effects of opioids can vary depending on the specific opioid [50], and might be explained by the various opioids' ability to activate different signalling pathways upon binding to the μ−opioid receptor on the cells of the immune system [51]. Moreover, one opioid does not exert the same effect on the level of the same cytokine [14,52], suggesting that the immunomodulatory properties of opioids are

**Table 3. Regression analysis between all the cytokines and symptom clusters. Cytokines are dependent variables, and symptom clusters and confounding factors (age, gender, BMI and smoking) independent variables.**

**SCL−90−r symptom cluster**

| Cytokine | Depression | | | Anxiety | | | Somatization | | | Psychoticism | | |
|---|---|---|---|---|---|---|---|---|---|---|---|---|
| | Coeff. | 95% CI | p−value | Coeff. | 95% CI | p−value | Coeff. | 95% CI | p−value | Coeff. | 95% CI | p−value |
| TNF−α | −0.03 | −.07; 0.05 | 0.821 | −0.05 | −0.08; 0.05 | 0.710 | 0.07 | −0.05; 0.09 | 0.578 | 0.01 | −0.10; 0.11 | 0.949 |
| IFN−γ | −0.11 | −0.24; 0.09 | 0.352 | −0.15 | −0.29; 0.07 | 0.242 | −0.08 | −0.25; 0.13 | 0.522 | −0.09 | −0.41; 0.18 | 0.447 |
| IL−1b | −0.07 | −0.13; 0.07 | 0.581 | −0.09 | −0.15; 0.07 | 0.462 | −0.04 | −0.13; 0.10 | 0.766 | −0.16 | −0.30; 0.06 | 0.182 |
| IL−1Ra | −0.11 | −0.09; 0.03 | 0.371 | −0.03 | −0.08; 0.06 | 0.789 | 0.01 | −0.07; 0.08 | 0.923 | 0.03 | −0.10; 0.13 | 0.799 |
| IL−2 | −0.12 | −0.11; 0.04 | 0.304 | −0.03 | −0.09; 0.07 | 0.818 | −0.02 | −0.09; 0.08 | 0.892 | 0.00 | −0.13; 0.14 | 0.978 |
| IL−4 | −0.17 | −0.13; 0.09 | 0.136 | −0.12 | −0.13; 0.04 | 0.321 | −0.11 | −0.13; 0.05 | 0.349 | 0.03 | −0.12; 0.16 | 0.820 |
| IL−6 | −0.09 | −0.13; 0.06 | 0.423 | −0.12 | −0.16; 0.05 | 0.321 | −0.10 | −0.16; 0.06 | 0.373 | −0.04 | −0.20; 0.14 | 0.724 |
| IL−8 | 0.04 | −0.06; 0.09 | 0.749 | −0.01 | −0.09; 0.08 | 0.954 | 0.04 | −0.07; 0.10 | 0.710 | 0.00 | −0.14; 0.14 | 0.998 |
| IL−9 | −0.18 | −0.24; 0.03 | 0.118 | −0.12 | −0.22; 0.07 | 0.312 | −0.08 | −0.21; 0.10 | 0.461 | −0.04 | −0.28; 0.20 | 0.721 |
| IL−13 | −0.09 | −0.12; 0.05 | 0.477 | −0.04 | −0.11; 0.09 | 0.778 | 0.01 | −0.09; 0.10 | 0.955 | −0.07 | −0.20; 0.11 | 0.551 |
| IL−15 | −0.02 | −0.17; 0.15 | 0.904 | 0.05 | −0.14; 0.21 | 0.663 | 0.03 | −0.16; 0.20 | 0.803 | 0.10 | −0.17; 0.40 | 0.412 |
| IL−17a | −0.06 | −0.10; 0.06 | 0.603 | −0.05 | −0.10; 0.07 | 0.659 | 0.11 | −0.05; 0.13 | 0.335 | 0.02 | −0.13; 0.15 | 0.865 |
| MCP−1 | 0.11 | −0.05; 0.14 | 0.366 | 0.03 | −0.10; 0.12 | 0.843 | 0.06 | −0.08; 0.14 | 0.593 | 0.13 | −0.07; 0.27 | 0.251 |
| IP−10 | −0.23 | −0.18; −0.00 | *0.044** | −0.12 | −0.15; 0.05 | 0.301 | −0.10 | −0.15; 0.06 | 0.391 | −0.08 | −0.22; 0.11 | 0.500 |
| Eotaxin | −0.05 | −0.08; 0.05 | 0.633 | −0.03 | −0.08; 0.06 | 0.770 | 0.06 | −0.05; 0.09 | 0.606 | 0.06 | −0.08; 0.15 | 0.576 |
| MIP−1 α | −0.02 | −0.08; 0.06 | 0.844 | 0.01 | −0.07; 0.08 | 0.910 | 0.13 | −0.03; 0.12 | 0.259 | 0.05 | −0.10; 0.15 | 0.643 |
| MIP−1 β | 0.07 | −0.02; 0.04 | 0.577 | 0.02 | −0.03; 0.04 | 0.850 | −0.01 | −0.04; 0.04 | 0.967 | 0.03 | −0.05; 0.06 | 0.817 |
| RANTES | 0.05 | −0.05; 0.09 | 0.662 | 0.06 | −0.06; 0.09 | 0.646 | −0.05 | −0.10; 0.06 | 0.646 | −0.02 | −0.14; 0.11 | 0.846 |
| GM−CSF | −0.25 | −0.19; −0.01 | *0.039** | −0.19 | −0.19; 0.02 | 0.117 | −0.12 | −0.16; 0.06 | 0.324 | −0.12 | −0.26; 0.09 | 0.325 |
| VEGF | 0.15 | −0.03; 0.13 | 0.199 | 0.12 | −0.05; 0.14 | 0.337 | 0.00 | −0.09; 0.09 | 0.976 | 0.02 | −0.13; 0.16 | 0.844 |

**SCL−90−r symptom cluster**

| Cytokine | Phobic anxiety | | | Paranoid ideation | | | Hostility | | | Interpersonal sensitivity | | | Obsessive−compulsive | | |
|---|---|---|---|---|---|---|---|---|---|---|---|---|---|---|---|
| | Coeff. | 95% CI | p−value | Coeff. | 95% CI | P−value | Coeff. | 95% CI | P−value | Coeff. | 95% CI | P−value | Coeff. | 95% CI | P−value |
| TNF−α | 0.057 | −0.05; 0.08 | 0.629 | −0.00 | −0.08; 0.07 | 0.981 | 0.16 | −0.03; 0.15 | 0.163 | 0.12 | −0.04; 0.13 | 0.290 | −0.01 | −0.06; 0.05 | 0.907 |
| IFN−γ | −0.184 | −0.31; 0.04 | 0.124 | −0.19 | −0.39; 0.04 | 0.104 | −0.18 | −0.35; 0.05 | 0.133 | −0.05 | −0.23; 0.14 | 0.652 | −0.04 | −0.19; 0.14 | 0.770 |
| IL−1b | −0.112 | −0.15; 0.06 | 0.351 | −0.17 | −0.22; 0.03 | 0.141 | −0.00 | −0.15; 0.14 | 0.970 | 0.05 | −0.10; 0.16 | 0.655 | −0.07 | −0.12; 0.07 | 0.581 |
| IL−1Ra | 0.162 | −0.02; 0.11 | 0.172 | −0.02 | −0.09; 0.07 | 0.859 | 0.15 | −0.03; 0.15 | 0.197 | 0.06 | −0.06; 0.10 | 0.611 | −0.08 | −0.08; 0.04 | 0.526 |
| IL−2 | 0.070 | −0.05; 0.10 | 0.546 | −0.05 | −0.12; 0.07 | 0.647 | 0.08 | −0.07; 0.14 | 0.486 | 0.07 | −0.07; 0.13 | 0.522 | −0.09 | −0.10; 0.04 | 0.447 |
| IL−4 | −0.015 | −0.09; 0.08 | 0.895 | −0.10 | −0.14; 0.06 | 0.382 | 0.04 | −0.09; 0.12 | 0.755 | 0.01 | −0.10; 0.10 | 0.961 | −0.14 | −0.12; 0.03 | 0.235 |
| IL−6 | −0.082 | −0.14; 0.07 | 0.488 | −0.10 | −0.17; 0.08 | 0.487 | 0.07 | −0.09; 0.18 | 0.527 | 0.12 | −0.06; 0.19 | 0.308 | −0.07 | −0.12; 0.07 | 0.547 |
| IL−8 | −0.032 | −0.10; 0.07 | 0.790 | −0.09 | −0.14; 0.06 | 0.415 | 0.07 | −0.07: 0.12 | 0.567 | 0.03 | −0.08; 0.10 | 0.822 | 0.01 | −0.07; 0.08 | 0.964 |
| IL−9 | 0.103 | −0.08; 0.20 | 0.370 | 0.02 | −0.16; 0.19 | 0.880 | 0.08 | −0.13; 0.27 | 0.492 | 0.08 | −0.12; 0.25 | 0.484 | −0.21 | −0.25; 0.01 | 0.063 |
| IL−13 | 0.069 | −0.06; 0.12 | 0.564 | −0.03 | −0.12; 0.10 | 0.817 | 0.16 | −0.04; 0.20 | 0.181 | 0.11 | −0.06; 0.17 | 0.342 | −0.12 | −0.13; 0.04 | 0.291 |
| IL−15 | 0.014 | −0.16; 0.18 | 0.909 | −0.02 | −0.22; 0.19 | 0.893 | 0.17 | −0.06; 0.39 | 0.152 | 0.06 | −0.16; 0.26 | 0.624 | −0.06 | −0.11; 0.19 | 0.607 |
| IL−17a | 0.059 | −0.06; 0.10 | 0.615 | 0.07 | −0.07; 0.13 | 0.525 | 0.20 | −0.01; 0.21 | 0.067 | 0.05 | −0.05; 0.15 | 0.343 | −0.04 | −0.09; 0.06 | 0.732 |
| MCP−1 | −0.017 | −0.11; 0.09 | 0.885 | 0.04 | −0.10; 0.15 | 0.720 | 0.17 | −0.04; 0.23 | 0.152 | 0.20 | −0.02; 0.23 | 0.084 | 0.16 | −0.03; 0.16 | 0.187 |
| IP−10 | −0.027 | −0.11; 0.09 | 0.821 | −0.13 | −0.19; 0.05 | 0.260 | −0.13 | −0.18; 0.05 | 0.244 | −0.15 | −0.18; 0.03 | 0.176 | −0.14 | −0.14; 0.04 | 0.242 |
| Eotaxin | 0.037 | −0.06; 0.08 | 0.738 | −0.05 | −0.10; 0.07 | 0.672 | 0.11 | −0.03; 0.14 | 0.129 | 0.05 | −0.06; 0.10 | 0.641 | 0.01 | −0.06; 0.07 | 0.952 |
| MIP−1 α | 0.039 | −0.06; 0.09 | 0.742 | 0.04 | −0.08; 0.10 | 0.756 | 0.15 | −0.03; 0.15 | 0.210 | 0.15 | −0.03; 0.14 | 0.194 | 0.06 | −0.05; 0.08 | 0.608 |
| MIP−1 β | −0.145 | −0.05; 0.01 | 0.220 | −0.08 | −0.06; 0.03 | 0.494 | −0.06 | −0.06; 0.03 | 0.643 | −0.00 | −0.04; 0.04 | 0.979 | 0.08 | −0.02; 0.04 | 0.495 |
| RANTES | −0.088 | −0.10; 0.05 | 0.461 | −0.07 | −0.12; 0.06 | 0.547 | −0.10 | −0.14; 0.05 | 0.391 | 0.06 | −0.07; 0.11 | 0.624 | 0.16 | −0.02; 0.11 | 0.188 |
| GM−CSF | −0.257 | −0.21; −0.01 | *0.031** | −0.21 | −0.23; 0.01 | 0.076 | −0.09 | −0.18; 0.08 | 0.441 | −0.12 | −0.18; 0.05 | 0.279 | −0.22 | −0.18; 0.01 | 0.063 |
| VEGF | −0.115 | −0.12; 0.04 | 0.337 | −0.08 | −0.14; 0.07 | 0.476 | 0.01 | −0.11; 0.12 | 0.914 | −0.02 | −0.12; 0.09 | 0.831 | 0.08 | −0.05; 0.11 | 0.487 |

**Table 4. Difference between non−OMT and OMT patients in demographic characteristics, virus load, psychiatric symptom−cluster and substance use last 30 days.**

| | | Non−OMT | OMT | p−value |
|---|---|---|---|---|
| | | N = 67 (56%) | N = 53 (44%) | |
| Demographics | | | | |
| Female gender | N (%) | 26 (39) | 13 (25) | 0.097 |
| Age (years) | Median (IQR) | 47 (37−56) | 40 (34−51) | *0.038* |
| On psychotropic drugs | N (%) | 15 (22) | 19 (35) | 0.236 |
| BMI | Median (IQR) | 25 (22−28) | 26 (23−30) | 0.421 |
| Smoking | N (%) | 39 (58) | 32 (61) | 0.140 |
| Norwegian nationality | N (%) | 60 (90) | 47 (89) | 0.462 |
| Virus load HCV RNA (IU/L*1000) | Median (IQR) | 1255 (435−3297) | 633 (276−3110) | 0.154 |
| Mental health scores (cluster) from SCL−90−R | | | | |
| Depression | Median (IQR) | 1.15 (0.54−1.77) | 1.23 (0.65−2.16) | 0.370 |
| Anxiety | Median (IQR) | 0.90 (0.30−1.70) | 1.10 (0.60−1.80) | 0.207 |
| Somatization | Median (IQR) | 1.25 (0.67−1.92) | 1.13 (0.52−1.83) | 0.588 |
| Psychoticism | Median (IQR) | 0.30 (0.10−0.70) | 0.40 (0.10−0.60) | 0.596 |
| Phobic anxiety | Median (IQR) | 0.43 (0.00−1.29) | 0.71 (0.14−1.14) | 0.569 |
| Paranoid ideation | Median (IQR) | 0.50 (0.17−2.37) | 0.33 (0.17−1.72) | 0.589 |
| Obsessive−compulsive | Median (IQR) | 1.20 (0.70−3.02) | 1.30 (0.79−3.13) | 0.628 |
| Interpersonal sensibility | Median (IQR) | 0.67 (0.33−1.22) | 0.56 (0.11−1.22) | 0.536 |
| Anger−hostility | Median (IQR) | 0.50 (0.17−0.83) | 0.33 (0.17−0.38) | 0.510 |
| Substance use last 30 days | | | | |
| Alcohol | N (%) | 35 (53) | 20 (43) | 0.272 |
| Cannabis | N (%) | 20 (33) | 21 (44) | 0.268 |
| Amphetamine | N (%) | 9 (17) | 4 (9) | 0.222 |
| Benzodiazepines | N (%) | 11 (26) | 17 (39) | 0.218 |
| Opioids | N (%) | 3 (8) | 1 (2) | 0.229 |
| Heroin | N (%) | 1 (3) | 1 (2) | 0.882 |

OMT, opioid maintenance treatment; BMI, body mass index; HCV, hepatitis C virus; SCL−90−R, Symptoms Check−List−90−Revised; IQR, interquartile range. Significant values (p<0.05) shown in bold.

subject to the milieu, such as concomitant infection [53] and duration of opioid treatment [13,23]. Additionally, opioids exert their immunomodulatory effects not only directly on cells of the immune system, but also indirectly through the hypothalamic−pituitary−adrenal axis [54,55]. Lastly, cytokines act in an autocrine and paracrine manner, complicating their interactions with opioids [56].

## Limitations

Our study recorded ongoing psychotropic medication dichotomously (yes/no) without accounting for the type or dosage. Different psychotropic drugs have various effects on cytokines [57,58], which may have influenced our results. However, the prevalence of ongoing psychotropic medication did not differ significantly between the non−OMT and OMT groups. The HCV−genotype was not taken into consideration, and HIV status was unknown, both of which are known to affect cytokine levels [59,60]. Many recordings were missing for smoking [61] and on−going administration of anti−inflammatory drugs [62] and might have been contributed to the results. The lack of a healthy control group limits comparisons and contextualization of cytokine levels. The cross−sectional design precludes examination of longitudinal changes in cytokines

**Table 5. Hierarchical linear regression analysis of the relationship between the cytokines IP−10 and GM−CSF and their relationships to SCL−90−R symptom clusters previously displaying significant association, stratified into non−OMT and OMT patients. Cytokines are dependant variable.**

| Depression/IP−10 | Non−OMT patients | | | OMT patients | | |
|---|---|---|---|---|---|---|
| | Coef. | 95% CI | p−value | Coef. | 95% CI | p−value |
| Adjusted for age | −0.051 | −0.118; 0.078 | 0.683 | −0.295 | −0.227; −0.012 | **0.030** |
| Adjusted for age and gender | −0.066 | −0.124; 0.072 | 0.601 | −0.337 | −0.242; −0.032 | **0.012** |
| Adjusted for age, gender and BMI | −0.017 | −0.108; 0.095 | 0.902 | −0.492 | −0.306; −0.076 | **0.002** |
| Adjusted for age, gender, BMI and smoking | −0.067 | −0.139; 0.089 | 0.658 | −0.489 | −0.295; −0.082 | **0.001** |
| Depression/GM−CSF | | | | | | |
| Adjusted for age | −0.143 | −0.170; 0.043 | 0.239 | −0.273 | −0.186; −0.005 | **0.040** |
| Adjusted for age and gender | −0.150 | −0.175; 0.042 | 0.224 | −0.328 | −0.200; −0.029 | **0.010** |
| Adjusted for age, gender and BMI | −0.087 | −0.156; 0.081 | 0.527 | −0.395 | −0.246; −0.033 | **0.012** |
| Adjusted for age, gender, BMI and smoking | −0.103 | −0.180; 0.086 | 0.479 | −0.366 | −0.239; −0.026 | **0.017** |
| Phobic anxiety/GM−CSF | | | | | | |
| Adjusted for age | −1.189 | −0.170; 0.043 | 0.239 | −2.108 | −0.186; −0.005 | **0.040** |
| Adjusted for age and gender | −0.176 | −0.171; 0.027 | 0.151 | −0.310 | −0.245; −0.031 | **0.013** |
| Adjusted for age, gender and BMI | −0.122 | −0.146; 0.055 | 0.367 | −0.346 | −0.293; −0.027 | **0.020** |
| Adjusted for age, gender, BMI and smoking | −0.160 | −0.181; 0.056 | 0.294 | −0.323 | −0.274; −0.014 | **0.031** |

OMT, opioid maintenance treatment; BMI, body mass index; HCV, hepatitis C virus; CI, confidence interval; Coef, β coefficient. Significant values (p<0.05) shown in bold.

levels linked to duration of opioid treatment, which may influence cytokine−symptom relationships [13,23]. Because of the number of cytokines analysed, there is a risk of type 1 error. Therefore, these data should be regarded as exploratory and hypothesis−generating, rather than definitive.

## Strengths

All interviews were conducted by the first author, reducing inter−rater variability [5]. The examination of 27 cytokines provides a comprehensive view of potential immune dysregulation, accounting for the complex interaction between cytokines [56]. We recorded data concerning illicit drug use during the last 30 days prior to blood sampling and found that there were no inter−group differences. This reduces the probability that polydrug use had a role in the differing cytokine levels [63–65]. The study design accounted for the high prevalence of HCV among OMT−patients [11], reducing confounding effects from infection. All cytokine measurements were performed by the same experienced technician using a standardized protocol, and blood samples were collected at consistent times, minimizing diurnal variation [66].

## Conclusions

This study investigated the association between 27 cytokines and self−reported psychiatric symptom clusters in a population with chronic hepatitis C virus infection. We found that only in the opioid−dependent group were IP−10 and GM−CSF negatively associated with depression, while GM−CSF was negatively associated with phobic anxiety. The negative associations may indicate that long−term opioid administration dampens the immunological response to psychiatric symptoms.

## Supporting information

**S1 Table.** Median cytokine values of healthy controls.
(DOCX)

## Acknowledgments

We sincerely thank Elin Malin for her invaluable contribution to the patient inclusion process. Our gratitude extends to Judith Krey Ludviksen and Evy Eide for conducting the cytokine analyses, and to Lill Magna Lekanger for her meticulous SPSS data preparation. Your efforts were essential to the success of this study.

## Author contributions

**Conceptualization:** Kristin Nygard−Odeh, Hedda Soløy−Nilsen, Magnhild Gangsøy−Kristiansen, Tom Eirik Mollnes.

**Data curation:** Kristin Nygard−Odeh, Ole Lars Brekke.

**Formal analysis:** Kristin Nygard−Odeh.

**Funding acquisition:** Kristin Nygard−Odeh.

**Investigation:** Kristin Nygard−Odeh.

**Methodology:** Kristin Nygard−Odeh, Hedda Soløy−Nilsen, Tom Eirik Mollnes, Michael Berk.

**Project administration:** Kristin Nygard−Odeh.

**Resources:** Ole Lars Brekke.

**Supervision:** Michael Berk.

**Visualization:** Kristin Nygard−Odeh.

**Writing – original draft:** Kristin Nygard−Odeh.

**Writing – review & editing:** Kristin Nygard−Odeh, Hedda Soløy−Nilsen, Magnhild Gangsøy−Kristiansen, Ole Lars Brekke, Tom Eirik Mollnes, Michael Berk.

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
