## [Decision Letter · Decision Letter 0]

Dear Dr. Nygard-Odeh,

Thank you for submitting your manuscript to PLOS ONE. After careful consideration, we feel that it has merit but does not fully meet PLOS ONE’s publication criteria as it currently stands. Therefore, we invite you to submit a revised version of the manuscript that addresses the points raised during the review process.

We look forward to receiving your revised manuscript.

Kind regards,

Jason T. Blackard, PhD

Academic Editor

PLOS ONE

“The study was in part funded by research grant from Northern Norway Regional health authority (reference: RUS1303-16).”

“NO.”

5. In the online submission form you indicate that your data is not available for proprietary reasons and have provided a contact point for accessing this data. Please note that your current contact point is a co-author on this manuscript. According to our Data Policy, the contact point must not be an author on the manuscript and must be an institutional contact, ideally not an individual. Please revise your data statement to a non-author institutional point of contact, such as a data access or ethics committee, and send this to us via return email. Please also include contact information for the third party organization, and please include the full citation of where the data can be found.

Additional Editor Comments:

This is a cross-sectional study evaluating cytokine expression in persons with HCV infection and psychiatric symptoms.

There are several awkward phrases . . . See examples (there are others) below.  The manuscript should be reviewed carefully by a native English speaker and/or a professional editing service.

Line 45-47:  awkward wording; rephrase this sentenceLine 48:  undue is not hyphenated

Line 65:  “. . . may thus not . . . “ should be reworded

Was HCV genotype considered?

What is the HIV status of the population?  

Reviewers' comments:

Reviewer's Responses to Questions

**Comments to the Author**

1. Is the manuscript technically sound, and do the data support the conclusions?

Reviewer #1: No

Reviewer #2: Yes

2. Has the statistical analysis been performed appropriately and rigorously?

Reviewer #1: Yes

Reviewer #2: I Don't Know

3. Have the authors made all data underlying the findings in their manuscript fully available?

Reviewer #1: Yes

Reviewer #2: Yes

4. Is the manuscript presented in an intelligible fashion and written in standard English?

Reviewer #1: Yes

Reviewer #2: Yes

Reviewer #1: In this article, the authors investigated the association between cytokines and psychiatric symptoms, as well as the relationship between opioids and cytokines in patients with chronic hepatitis C virus infection, of whom nearly 44% are on opioid maintenance treatment (OMT). This cross-sectional study included 120 outpatients referred for antiviral hepatitis C treatment. Serum levels of 27 cytokines were measured using a multiplexed assay, and self-reported psychiatric symptom clusters were assessed with the Symptoms Check-List-90-R. The data showed that, after correcting for multiple confounding factors, IP-10 and GM-CSF were negatively associated with depression, while GM-CSF was also negatively associated with phobic anxiety; these associations were found to be opioid-dependent. The authors conclude that the immunomodulatory qualities of opioids are diverse and likely influenced by both intrinsic and extrinsic factors, the complexity of which is yet to be fully understood. Although this study provides a link between psychiatric disorders and inflammation in the context of opioid use, several concerns may temper enthusiasm for the outcomes.

One of the major issues with substance abuse-related studies is polydrug use. It is unclear whether the authors considered this issue while interpreting the results.

The type of opioids used may impact the study outcomes. Is there any information about the participants' opioid use history?

The duration of OMT is an important factor to consider when interpreting the data.

Information comparing individual cytokine levels between OMT and non-OMT groups is lacking. It would be beneficial to compare these groups to determine whether OMT or opioid use significantly affects cytokine levels.

The relevance of these findings in the context of inflammation and substance use is not discussed.

The conclusion stated, “The immunomodulatory qualities of opioids are diverse and probably influenced by both intrinsic and extrinsic factors, the complexity of which is yet to be fully understood,” is vague and not supported by the current data.

Most of the cytokines measured are very low or near the lower limit of detection (LLOD), and the interquartile range (IQR) data are very close. Did the authors compare the data with a control group to validate the detection limits of the assay?

Reviewer #2: This is an interesting paper on association of cytokine levels in people with HCV +/- opioid use. This is an interesting question with significance for the neurotropic effects of other infections both alone and in the context of substance use. Overall the paper is well written and the conclusions are supported by the data. The strength of the patient population and the importance of the findings support publication of what is essentially a single data set.

However, this work could be strengthened in a few notable ways using the data already present. The authors start by looking at the association of cytokine expression with different psychiatric states. They identify two cytokines and then look at what effect opioids have on their expression. I think analysis looking at association of ALL the cytokines in opioid negative vs opioid using would be valuable. That the IP10 and GMCSF changes are more significant in the opioid using population and non-existent in the opioid negative support that other cytokines may be affected, but not seen in the overall analysis. Given the relationship between both inflammation AND drug seeking behavior in females vs males an examination of gender as a variable is also needed. Obviously, these comparisons lower the power as the groups will be smaller. This leads to my next suggestion which is a discussion of the power of the study for each comparison made such that the reader can understand what may not be detected.

There also seems to be an opportunity to compare to HCV negative individuals using the data set from the authors previous works (reference 23).

**Do you want your identity to be public for this peer review?** For information about this choice, including consent withdrawal, please see our Privacy Policy

Reviewer #1: No

Reviewer #2: **Yes: ** Zachary Klase

---

## [Author Response · Author response to Decision Letter 1]

18 Feb 2025

Dear esteemed Academic editor and Reviewers,

By this response, the online submitted revised manuscript and through the online submission forms and procedures, we hope that all the editorial and revisional requirements have been met. The entire manuscript has been through an extensive language revision. In the “Revised manuscript with track changes”, additions in response to the editor’s and reviewer’s input are highlighted in yellow, and language edits are illustrated in red from the tracking.

Below are the responses to “Additional Editor Comments, and “Review comments to the Author”.

Additional editor comments:

We are grateful for your valuable input and your comments addressed below.

1. Was HCV genotype considered?

HCV genotype was not considered, but is of importance. It has been added as a point under “limitations” in the revised manuscript lines 267-268.

2. What is the HIV status of the population?

HIV-status was not investigated, and therefore unknown. This information has now been added in the revised manuscript lines 267-268.

Reviewer 1:

Thank you for your thorough and concise review of our paper. We hope that we have answered all your concerns below, and that the responses are to your satisfaction.

1. One of the major issues with substance abuse-related studies is polydrug use. It is unclear whether the authors considered this issue while interpreting the results.

Thank you for pointing out the lack of consideration of the impact polydrug use had on cytokine levels. The issue is addressed in the discussion of the revised manuscript lines 279-281.

2. The type of opioids used may impact the study outcomes. Is there any information about the participants' opioid use history?

Information pertaining to opioid use history (other than OMT-opioids) was limited to use in the last 30 days before inclusion and blood-sampling. This is illustrated in table 4 in the manuscript.

3. The duration of OMT is an important factor to consider when interpreting the data.

We agree that this is an important factor to consider and is now commented on in lines 271-273 in the revised manuscript.

4. The relevance of these findings in the context of inflammation and substance use is not discussed.

Placing our findings in the context of inflammation and substance use is of great relevance, and we have added a section in lines 251-254 in the manuscript.

5. The conclusion stated, “The immunomodulatory qualities of opioids are diverse and probably influenced by both intrinsic and extrinsic factors, the complexity of which is yet to be fully understood,” is vague and not supported by the current data.

We agree that the conclusion was vague and unsubstantiated by the data, and have revised and corrected it accordingly.

6. Most of the cytokines measured are very low or near the lower limit of detection (LLOD), and the interquartile range (IQR) data are very close. Did the authors compare the data with a control group to validate the detection limits of the assay.

We agree that that most of the cytokine values in the study participants were low and near LLOD and that the IQR data were very close. We did compare these data with a group of healthy controls (Supplementary table 1), and found the same there. The main reason for the variation in the lower limit of detection was that the lowest point of the standard curve varied slightly on different days. This is now commented in lines 170-184 including the addition of the supplementary table.

Reviewer 1 and 2:

Both Reviewers have commented on the value of stratifying the study patients and comparing cytokine levels between the respective groups from the get-go, rather than the two with significant association to psychiatric symptoms in the entire study sample. While we agree with the reviewers’ statement that this would have been an interesting angle, the aim of this study was to characterize associations between cytokines and mental health symptoms in HCV patients. It was therefore beyond the scope of this report.

Reviewer 2:

We greatly appreciate your assessment, and hope that the answers below are to your satisfaction.

1. Given the relationship between both inflammation AND drug seeking behaviour in females vs males an examination of gender as a variable is also needed

Gender was taken into consideration, and included as an independent variable in the regression analysis.

2. This leads to my next suggestion which is a discussion of the power of the study for each comparison made such that the reader can understand what may not be detected.

Secondary outcomes were exploratory, and a priori power calculations for secondary outcomes were not conducted.

3. There also seems to be an opportunity to compare to HCV negative individuals using the data set from the authors previous works

The use of the patient sample from the work referred to, would undoubtedly have added valuable strength to our study. However, these are two separate projects, each with individual approvals from the Regional ethics committee. Comparing data across the two projects was therefore not something we had approval to do.

Sincerely, on behalf of the group

Kristin Nygård-Odeh

Corresponding author

---

## [Decision Letter · Decision Letter 1]

Cytokines interferon-γ- inducible protein 10 and granulocyte-macrophage colony-stimulating factor are associated with psychiatric symptoms in opioid-dependent patients: A cross- sectional study

PONE-D-24-48041R1

Dear Dr. Nygard-Odeh,

We’re pleased to inform you that your manuscript has been judged scientifically suitable for publication and will be formally accepted for publication once it meets all outstanding technical requirements.

Kind regards,

Jason T. Blackard, PhD

Academic Editor

PLOS ONE

Additional Editor Comments (optional):

None

Reviewers' comments:

Reviewer's Responses to Questions

**Comments to the Author**

Reviewer #1: All comments have been addressed

2. Is the manuscript technically sound, and do the data support the conclusions?

Reviewer #1: Yes

3. Has the statistical analysis been performed appropriately and rigorously?

Reviewer #1: Yes

4. Have the authors made all data underlying the findings in their manuscript fully available?

Reviewer #1: Yes

5. Is the manuscript presented in an intelligible fashion and written in standard English?

Reviewer #1: Yes

Reviewer #1: (No Response)

**Do you want your identity to be public for this peer review?** For information about this choice, including consent withdrawal, please see our Privacy Policy

Reviewer #1: No

---

## [Editor Report · Acceptance letter]

PONE-D-24-48041R1

PLOS ONE

Dear Dr. Nygard-Odeh,

I'm pleased to inform you that your manuscript has been deemed suitable for publication in PLOS ONE. Congratulations! Your manuscript is now being handed over to our production team.

Kind regards,

on behalf of

Dr. Jason T. Blackard

Academic Editor

PLOS ONE